# The Effect of Surface Oil on Ocean Wind Stress

Daneisha Blair [1], Yangxing Zheng [2],* and Mark A. Bourassa [1,2]

1   Department of Earth, Ocean, Atmospheric Science, Florida State University, Tallahassee, FL 32306, USA;
    db18g@fsu.edu (D.B.); bourassa@coaps.fsu.edu (M.A.B.)
2   Center for Ocean-Atmospheric Prediction Studies, Florida State University, Tallahassee, FL 32306, USA
*   Correspondence: yzheng@fsu.edu; Tel.: +1-850-644-1159

**Abstract:** This study provides, to the best of our knowledge, the first detailed analysis of how surface oil modifies air–sea interactions in a two-way coupled model, i.e., the coupled–ocean–atmosphere–wave–sediment–transport (COAWST) model, modified to account for oil-related changes in air–sea fluxes. This study investigates the effects of oil on surface roughness, surface wind, surface and near-surface temperature differences, and boundary-layer stability and how those conditions ultimately affect surface stress. We first conducted twin-coupled modeling simulations with and without the influence of oil over the Deepwater Horizon (DWH) oil spill period (20 April to 5 May 2010) in the Gulf of Mexico. Then, we compared the results by using a modularized flux model with parameterizations selected to match those selected in the coupled model adapted to either ignore or account for different atmospheric/oceanic processes in calculating surface stress. When non-oil inputs to the bulk formula were treated as being unchanged by oil, the surface stress changes were always negative because of oil-related dampening of the surface roughness alone. However, the oil-related changes to 10 m wind speeds and boundary-layer stability were found to play a dominant role in surface stress changes relative to those due to the oil-related surface roughness changes, highlighting that most of the changes in surface stress were due to oil-related changes in wind speed and boundary-layer stability. Finally, the oil-related changes in surface stress due to the combined oil-related changes in surface roughness, surface wind, and boundary-layer stability were not large enough to have a major impact on the surface current and surface oil transport, indicating that the feedback from the surface oil to the surface oil movement itself is insignificant in forecasting surface oil transport unless the fractional oil coverage is much larger than the value found in this study.

**Keywords:** COAWST modeling; surface oil; ocean surface stress; surface roughness; atmospheric-boundary-layer stability; surface wind

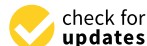



## 1. Introduction

The largest accidental marine oil spill in the history of the petroleum industry happened on 20 April 2010 in the Gulf of Mexico [1]. The Deepwater Horizon (DWH) oil platform suffered a catastrophic explosion that caused 11 deaths and injured 17 others [2]. The oil rig accident was estimated to have spilled 4.2 million barrels of oil over 87 days before the wellhead was finally capped on 15 July 2010 [3]. When oil first reaches the surface, gravity causes the oil to spread rapidly (relative to advection via currents) [4] in a layer that floats on top of the ocean; then, it is transported by winds and currents. That oil slick posed an unprecedented threat to Gulf of Mexico marine life as well as wetland and estuarine animal habitats [4,5]. Tracking the oil spill both on the surface and in the depths was necessary for planning mitigation efforts [6]. Trajectory forecasts using numerical models are one of the most important methods of tracking spills [7,8]. Many numerical ocean circulation models from different institutions were used for trajectory forecasting, such as the West Florida Shelf model [9], the Global Hybrid Coordinate Ocean Model [10], the Gulf of Mexico HYCOM (http://www.hycom.org, accessed on 1 May 2023), the South Atlantic Bight–Gulf of Mexico Model [11], the Real-Time Ocean Forecast System for the

North Atlantic Ocean [12], and the Intra-Americas Sea Nowcast/Forecast System [13]. Despite some success with these modeling activities, limitations on these modeling systems necessitate several future improvements [14]. For example, none of these models incorporated Stokes drift. They also did not consider the biological consumption of oil or the physical–chemical weathering processes.

There are several ways that surface oil can modify air–sea interactions (Figure 1). First, surface oil can change surface roughness by suppressing small gravity and capillary waves, leading to changes in friction velocity, ultimately contributing to a change in surface stress and surface current in the absence of other air–sea interaction process changes; as we know, friction velocity is expressed as

$$\mathbf{u}_* = \frac{k_v(\bar{\mathbf{u}}(z) - \bar{\mathbf{u}}_{\text{sfc}})}{\left[ln\left(\frac{z}{z_{om}} + 1\right) - \psi_m(z, z_{om}, L)\right]},$$

where $\bar{\mathbf{u}}(z)$ is the wind velocity at height $z$ (10 m in this case); $\bar{\mathbf{u}}_{\text{sfc}}$ is the velocity of the surface ocean current; $k_v$ is von Karman's constant; $z_{om}$ is the roughness length for momentum; $\psi_m$ is a function of boundary-layer stability for momentum; and $L$ is the Monin–Obukhov scale length.

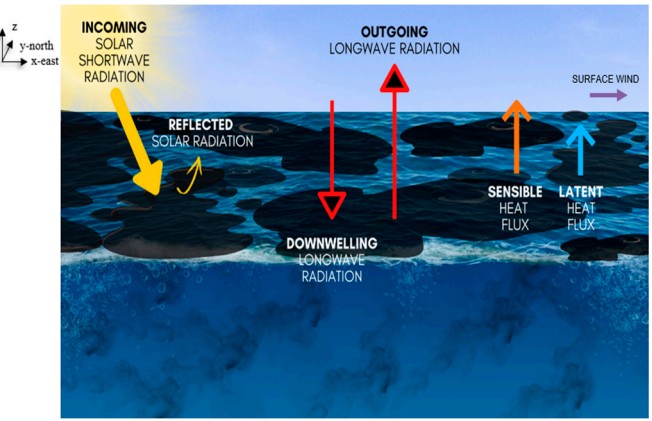

**Figure 1.** Three-dimensional schematic diagram depicting the ocean energy budget induced by an oil slick. The oil (brown shading) absorbs some of the incoming solar radiation (yellow arrow) and becomes warmer than the surrounding water. The increase in the surface oil temperature leads to an increase in outgoing longwave radiation (upward red arrow) and sensible heat flux (orange arrow). The smooth surface of the oil slick reduces friction velocity caused by surface winds at 10 m above the sea surface (purple arrow). Friction velocity is proportional to the energy lost through evaporation; therefore, the surface oil reduces evaporation, which, in turn, reduces the latent heat flux (blue arrow).

Second, surface oil can modify the sea surface temperature (SST) due to its influence on the ocean's near-surface energy budget. Latent and sensible heat fluxes can be changed because of the changes in friction velocity and temperature. If an oil slick is treated as an impermeable layer/membrane, then water vapor cannot pass through this layer, leaving only the oil-free portion of the surface available for the evaporation of water, thus reducing latent heat flux and an increase in the SST. Third, the SST changes due to the discrepancy between oil and water can also modify upward longwave radiative emissions [15]. The oil-related changes in the SST due to the ocean energy budget can then alter the atmospheric-boundary-layer stability, ultimately affecting the surface stress. None of the previous models considered how the discrepancy between water and oil in surface roughness, surface heat flux, and boundary-layer stability could modify surface stress on the ocean surface, which could impact the oil movement. Therefore, it is critically important to examine how oil modifies surface stress feedback and model surface oil movement in a two-way coupled model. To quantitatively assess how surface oil could

change surface stress, one should understand how oil changes surface roughness, surface wind, and boundary-layer stability. We expected that surface stress will change because of oil impacts on the surface roughness, surface temperature, and atmospheric boundary layer and that the changed stress will modify Ekman surface currents.

This study aims to produce a more realistic physical environment for examining the impact of surface oil on surface stress and surface oil transport using a two-way coupled modeling system. The two-way coupling allows different physical processes of the atmosphere, ocean, and waves to interact with each other [16–18]. To this end, a high-resolution two-way coupled–ocean–atmosphere–wave–sediment–transport (COAWST) modeling system [19] for the Gulf of Mexico region was utilized over the DWH oil spill period. To the best of our knowledge, this is the first time a two-way coupled model was used to study surface oil transport where oil itself modifies the coupling processes. This study was motivated by the limitations of the above-mentioned oil trajectory numerical models and by our previous study [7], which showed (assuming a water surface covered with oil) that there were likely to be substantial changes in oil transport due to oil-induced changes in the SST gradient, surface roughness, and surface stress.

The remainder of this paper is organized as follows. Section 2 includes a description of the models. Section 3 describes the methodology. Section 4 presents the results concerning the effect of surface oil on surface stress due to oil-related changes in surface roughness, surface winds, and atmospheric-boundary-layer stability separately. Section 5 discusses some caveats, and Section 6 summarizes the results.

## 2. COAWST Modeling System

### 2.1. Model Components

The COAWST modeling system [19] comprises several components, including models for the ocean, the atmosphere, sea surface waves, sediment transport, and a coupler. The Model Coupling Toolkit (MCT) allows for exchanges of prognostic fields between the various components, as shown in Figure 2.

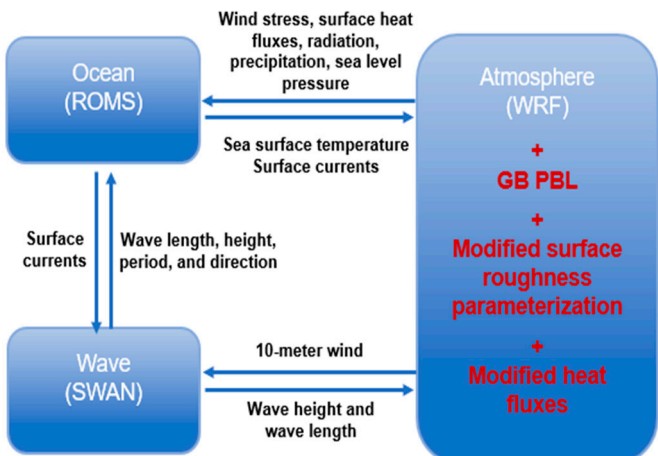

**Figure 2.** Coupled–ocean–atmosphere–wave–sediment–transport (COAWST) configuration and exchanged data fields. The COAWST coupling was modified to include the changes in physics parameterizations to support our oil spill simulation.

The COAWST's ocean model is the Regional Ocean Modeling System (ROMS). ROMS is a free-surface, terrain-following numerical model that solves the three-dimensional Reynolds-averaged Navier–Stokes equations using hydrostatic and Boussinesq approximations [20,21]. The COAWST's atmospheric model component is the Weather Research and Forecasting Model (WRF) [22]. This is a nonhydrostatic, quasi-compressible atmospheric model with boundary-layer physics schemes and a variety of physical parameterizations of sub-grid scale processes for predicting meso- and microscales of motions. The WRF

incorporates various physical processes including microphysics, cumulus parameterization, longwave and shortwave radiation, surface layer, land surface, and the planetary boundary layer (PBL). In most cases, there are multiple options for these parameterizations; thus, the physics options considered in our study are listed in Table 1. Model physics parameters were chosen to best support the DWH oil spill development and promote an accurate simulation. In our COAWST modeling system, modifications of surface roughness length and surface moisture fluxes were used in the WRF module to model a surface oil spill's transport. Section 2.2 describes the modification of the surface roughness length and the modification of the surface moisture flux used in the COAWST system. The wave model is called Simulating WAves Nearshore (SWAN). SWAN is a spectral wave model specifically designed for shallow water that solves the wave action balance equation in both spatial and spectral spaces [23]. SWAN simulates wind–wave generation and propagation in coastal waters considering various sources and sink terms, including refraction, diffraction, shoaling, wave–wave interactions, and dissipation terms [19].

**Table 1.** Physical parameterizations used in this study.

| Parameterization | Physics Options | References |
|---|---|---|
| Microphysics | Thompson graupel scheme | [24] |
| cumulus | Grell–Freitas ensemble scheme | [25] |
| Longwave/shortwave radiation | RRTMG scheme | [26] |
| Surface layer | Eta similarity scheme | [27] |
| Land surface | Unified Noah land surface model | [28] |
| Planetary boundary layer | GB scheme | [29] |

*2.2. Parameterization Modifications*

2.2.1. Surface Roughness Length

One of the physical processes that was not considered in the previously mentioned forecast models is that oil changes surface roughness, which can affect surface stress and oil transport, as suggested by an idealized study [7]. Herein, an innovative momentum roughness length parameterization is embedded in the WRF model to solve the surface water roughness and surface oil roughness. The new momentum roughness length parameterization is a modification of the Bourassa–Vincent–Wood flux model (BVW) [30,31], as applied in [7], which describes three types of surface features contributing to surface roughness: an aerodynamically smooth surface, capillary waves, and gravity waves. The surface roughness length parameterization includes a mix of roughness values from surfaces with and without oil (tuned to the DWH spill):

$$z_{0m_i} = \beta_s \frac{0.11v}{|u_{*_i}|} + \left[\left(\left((1 - A_{oil})\beta_{c,water} + \varepsilon A_{oil}\beta_{c,oil}\right)\frac{b\sigma}{\rho_w|\mathbf{u}_*||\mathbf{u}_*\cdot\mathbf{e_i}|}\right)^2 + \left(\left((1 - A_{oil})\beta_{g,water} + \varepsilon A_{oil}\beta_{g,oil}\right)\frac{a|\mathbf{u}_*||\mathbf{u}_*\cdot\mathbf{e_i}|}{g}\right)^2\right]^{0.5} \quad (1)$$

where $\beta_s$, $\beta_{c,water}$, $\beta_{c,oil}$, $\beta_{g,water}$ and $\beta_{g,oil}$ are weights for the roughness length associated with an aerodynamically smooth surface, capillary waves, and gravity waves for water and oil [30]; $v$ is the molecular viscosity of air; $A_{oil}$ is the fractional coverage by oil; $\varepsilon$ represents the oil-dampening effects on capillary waves and short gravity waves, and $\varepsilon$ is set to 0.25 for oil dampening and 1 for oil-free water; $b = 0.019$ is a dimensionless constant for the capillary wave roughness length, which is determined from laboratory observations [31]; $\sigma$ is surface tension, which is determined via the sea surface temperature; $\rho_w$ is the water density; $\mathbf{u}_*$ is the friction velocity; $a = 0.035$ is Charnock's constant; and $g$ is gravitational acceleration [30]. The roughness length is anisotropic, with unit vectors parallel, $\mathbf{e}_1$, and perpendicular, $\mathbf{e}_2$, to the mean direction of wave motions. The value of $\beta_s$ is determined by the relationship $\beta_s = 1 - \left(\varepsilon A_{oil}\beta_{g,oil} + (1 - A_{oil})\beta_{g,water}\right)$. The roughness associated with capillary and

gravity waves was modified to be a weighted average of the terms for a water surface and an oil-covered surface. The new value of the weighted average terms represents an average over enough space/time for a smooth transition from an aerodynamically smooth surface to a rough surface.

For the water surface, $\beta_{c,water} = 0$ and $\beta_{g,water} = 0$ for $U_{eff} < U_{lim}$. For $U_{eff} \geq U_{lim}$

$$\beta_{c,water} = \tanh\left(0.4\left(U_{eff} - U_{lim}\right)^3\right) \tag{2}$$

$$\beta_{g,water} = \tanh\left(0.2\left(U_{eff} - U_{lim}\right)^3\right) \tag{3}$$

For the oil surface, $\beta_{c,oil} = 0$ and $\beta_{g,oil} = 0$ for $U_{eff} < U_{lim}$. For $U_{eff} \geq U_{lim}$

$$\beta_{c,oil} = \tanh\left(0.4\left(U_{eff} - U_{lim}\right)^3\right) \tag{4}$$

$$\beta_{g,oil} = \tanh\left(0.3\left(U_{eff} - U_{lim}\right)^3\right) \tag{5}$$

where $U_{eff} = u_*\left[\ln\left(\frac{z}{z_0} + 1\right) + \varphi(z, z_0, L)\right]/k$, $\psi(z, z_0, L)$ is a stability-dependent modifier, and $k$ and $L$ are von Kármán's constant and the Monin–Obukhov stability length, respectively. For the water surface, $U_{lim} = 1.0$ ms$^{-1}$, and for the Deepwater Horizon oil-covered region, the value of $U_{lim} = 7.0$ ms$^{-1}$. These values of $U_{lim}$ for oil were estimated based on scatterometer observations of the surface roughness associated with the DWH spill. Using Equations (1)–(5) is good for estimating surface roughness length because this approximation is qualitatively consistent with the wave spectra measurement of Cox and Munk [32] in the sense that both gravity waves and capillary waves are suppressed by oil, and this dampening is parameterized in a manner that can be easily included in the boundary-layer model. It should be pointed out that the value of $U_{lim}$ is not general and will vary greatly depending on the oil conditions. Further work could be performed to better estimate the values of $U_{lim}$ and $\varepsilon$; however, this is beyond the scope of this study. Here, our goal is to show that these considerations are worthy of further analysis for modeling boundary layers and oil spill trajectories for large oil spills.

The new surface roughness length parameterization (1) assumes that the surface Is either oil-free, totally covered with oil, or partially covered with oil. For partial coverage, we modified the roughness length parameterization to include a contribution from capillary waves that are dampened ($0 < A_{oil} < 1$) and an oil-free portion ($1 - A_{oil}$) where they are not dampened. In Section 2.4, we will describe how we tuned $A_{oil}$ for much more general use in the COAWST system.

### 2.2.2. Surface Stress Flux and Latent and Sensible Heat Fluxes

The fluxes of stress ($\tau$), latent heat ($E$), and sensible heat ($H$) follow.

$$\boldsymbol{\tau} = \rho|\mathbf{u}_*|\mathbf{u}_* \tag{6}$$

$$E = (1 - A_{oil})\rho L_v|\mathbf{u}_*|q_* \tag{7}$$

$$H = \rho C_p|\mathbf{u}_*|\theta_* \tag{8}$$

where $\rho$ is the density of air, $A_{oil}$ is the fractional coverage by oil, $L_v$ is the latent heat of vaporization, and $C_p$ is the heat capacity of air. $q_*$ and $\theta_*$ are analogous to $\mathbf{u}_*$; however, they apply to humidity and potential temperature, which are determined by

$$\bar{\mathbf{u}}(z) - \bar{\mathbf{u}}(sfc) = \frac{\mathbf{u}_*}{k_v}\left[\ln\left(\frac{z}{z_{0m}} + 1\right) - \psi_m(z, z_{0m}, L)\right] \tag{9}$$

$$\bar{\theta}(sfc) - \bar{\theta}(z) = \frac{\theta_*}{k_v}\left[\ln\left(\frac{z}{z_{0\theta}} + 1\right) - \psi_\theta(z, z_{0\theta}, L)\right] \tag{10}$$

$$\bar{q}(sfc) - \bar{q}(z) = \frac{q_*}{k_v}\left[\ln\left(\frac{z}{z_{0q}} + 1\right) - \psi_q(z, z_{0q}, L)\right] \tag{11}$$

where the overbar indicates a mean, the boundary-layer stability terms for momentum, heat, and moisture (i.e., $\psi_m$, $\psi_\theta$, $\psi_q$) are calculated with an Obukhov stability length scale [27], and $\bar{\mathbf{u}}$, $\bar{\theta}$ and $\bar{q}$ at the surface and height (z) are obtained from the coupled model surface datasets. The stability terms are modified by air–sea temperature differences (and, to a much smaller extent, humidity differences) and stress, which are two paths through which the oil modification of the surface can influence heat and momentum fluxes in addition to the air–sea difference in the bulk formulas (Equations (9)–(11)). These equations allow us to examine the impacts of boundary-layer stability changes related to oil, of which the change in stress directly impacts the Ekman transport (wind-driven currents) of oil. Momentum roughness length parameterization considers the effects of capillary and gravity waves (Equations (1)–(5)), while temperature and moisture roughness length parameterizations consider the aerodynamically smooth surface only. Stability adjustment parameterization is determined by [33] for unstable conditions and by [34] for stable conditions.

### 2.3. Experiment Design

We conducted the twin experiments over the DWH period from 20 April to 5 May 2010. The model simulations were initialized to start 2 h after the DWH blowout on 20 April 2010, with the model output stored every 6 h. The WRF made simulations over the Gulf of Mexico (Figure 3), and its horizontal resolution was 5 km × 5 km. The WRF's initial and boundary conditions were obtained from the six-hourly Climate Forecast System Reanalysis (CFSR) data, which were produced by National Centers for Environmental Prediction (NCEP).

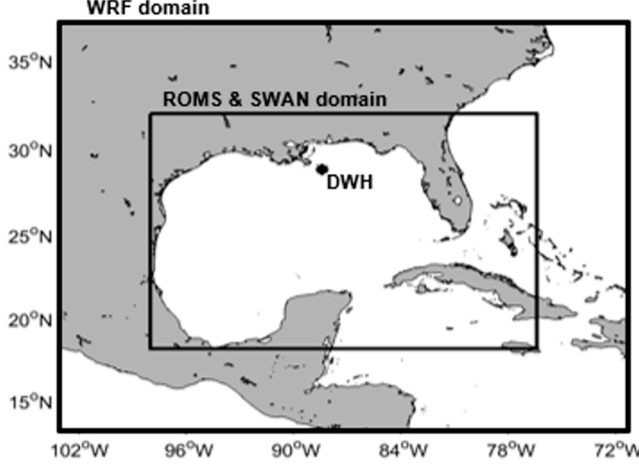

**Figure 3.** The model domains used for the COAWST simulations.

The ROMS domains (2 km × 2 km resolution) are within the bounds of the corresponding WRF domains for stability. Since the WRF and ROMS grids are not co-located, the Spherical Coordinate Remapping Interpolation Package (SCRIP) computes interpolation weights between WRF and ROMS. ROMS is initialized with HYCOM/NCODA (Hybrid

Coordinate Model/NRL-Coupled Ocean Data Assimilation Data) 1/12° reanalysis to reduce the spin-up time. The lower resolution in the WRF will reduce the magnitude of feedback relative to a model with a 2 km resolution in all domains. This was a necessary tradeoff given the computational requirements and the available resources and is a common practice when modeling is focused on the ocean [35].

SWAN simulations were performed on the same grid as ROMS. The SWAN model was initialized from a quasi-stationary state calculated using North American Mesoscale Model (NAM) winds. The spin-up period required to generate the wind waves is typically 12 to 18 h [36]. The boundary conditions for SWAN, such as significant wave height, peak period, and average direction at the peak period, which were updated every 3 h, were specified every 20 km (at every tenth grid point, called spec points) on the lateral open boundaries by the 10-arc-minute Northern Atlantic regional grid WaveWatch III model output.

### 2.4. Assumptions for Oil Slick Simulation

In this study, the goal of the oil simulation was to mimic the DWH blowout where oil continued to reach the surface until the oil wellhead was sealed off. This version of our model does not contain the physics for oil buoyancy, which could allow the oil to float from the underground well to the surface; therefore, we injected oil as a passive tracer at the surface. Using oil as a passive tracer in the model allows us to easily incorporate the surface roughness and evaporation flux equations for oil at the surface of the DWH site. A passive tracer in this scenario means that the surface water in the DWH site is acting as surface oil. This is a common approach to modeling oil transport as a mature slick [37–41].

The fractional oil coverage in the DWH area was calculated based on the oil discharge, which is the wellhead discharge rate at 9900 m$^3$ per day, and by assuming that half of the oil reached the surface [3]. Following [4], it was assumed that the oil slick can initially be treated as distinct from water and that it spreads as a density flow on the surface. This initial expansion is very rapid (less than an hour) based on observations of natural slicks [4]. After this expansion, the initial surface slick is treated as being 50 km in diameter. We treated the oil density as crude oil (881 kg m$^{-3}$) and ignored changes in this density associated with the evaporation of the more volatile components of oil and weathering. Furthermore, we assumed that the rate of oil reaching the surface was constant throughout our modeling period. These assumptions were used to calculate an oil flux density added to the surface over the 50 km diameter circular area centered on the DWH site. While this is not a sophisticated plume model, such as in [42], it is suitable for the goals of this study. The results of this study are dependent on the effectiveness of our tuning of the oil coverage ($A_{oil}$).

The rate at which oil reaches the surface is determined as follows. The assumption is that 50% of the oil reaches the surface, combined with a discharge rate of oil of $9900 \frac{m^3}{day} \times 881 \frac{kg}{m^3} \times 50\% = 50.46 \frac{kg}{s}$. The flux density of oil reaching the surface is determined by dividing the above flux by the area (a 50 km diameter) in which oil is added to the surface, which obtains a surface oil flux density of $\frac{50.46 \frac{kg}{s}}{1.96 \times 10^9 m^2} = 2.57 \times 10^{-8} \frac{kg}{m^2 s}$. $A_{oil}$ was taken to be a linear function of the surface oil tracer concentration (SOTC) from SOTC = $0 \frac{kg}{m^3}$ and $A_{oil} = 0$, up to a SOTC value (hereafter, SOTC_crit), and $A_{oil} = 1$ when SOTC $\geq$ SOTC_crit. A crude preliminary estimate of SOTC_crit was made by running the model for 2 days with a completely passive oil tracer and taking the maximum resulting concentration to correspond to 10% oil coverage. The model was then run for 13.75 days with a preliminary SOTC_crit estimate of $2.5 \times 10^{-3} \frac{kg}{m^3}$ and a fully active oil tracer, affecting the surface roughness and surface heat flux. This model run yielded reasonable maximum values of $A_{oil}$, ranging from 0.1 to 0.3 over the course of the model run. For the final tuning of the model, the ocean heat budget was adjusted by changing the value of SOTC_crit and, hence, $A_{oil}$ until sea surface temperatures were consistent with observations achieved within and around the DWH oil slick, as measured by the ten channels of the Advanced Microwave Scanning Radiometer–Earth (AMSR-E)-observing satellite [43]. Due to frequent cloud cover in the area of interest, it was difficult to capture/locate the oil spill boundaries

through satellites (only 25 and 27 April 2010 had clear skies, which were the dates closest to the 20 April DWH blowout). Therefore, there were few pairs of temperatures inside and outside of the slick, and there was considerable (and difficult-to-quantify) uncertainty in this value. Furthermore, since the results were sensitive to the values of $A_{oil}$ and SOTC_crit (Sections 4.3 and 5), we recommend further study of the value of $A_{oil}$ for DWH and other oil spills.

## 3. Methodology

### 3.1. Flux Model for Oil

The above twin-coupled modeling experiments computed the surface stress with and without the influence of oil. These differences are due to the combined oil-related changes in surface roughness, surface wind, and atmospheric stability (including changes in air temperature and humidity). However, we still do not know which of these changes contributes most to surface stress changes. To explore how oil-related changes in surface roughness, surface wind, and boundary-layer stability impact the changes in surface stress separately, we use the coupled model output variables (i.e., SST, surface wind, surface air temperature, etc.) with and without the influence of oil as input variables in the Modularized Flux Testbed (MFT version 2021.1 [31]) to calculate oil-related changes in surface stress caused by each of these conditions. The MFT allows users to combine a wide range of parameterizations related to air–sea interactions and output many boundary-layer-related variables such as surface stress, sensible heat, latent heat, atmospheric-boundary-layer stability, and surface roughness. MFT physics parameterizations were selected to match those in the COAWST modeling system to ensure consistent surface stress calculations with output surface stress calculations from the modeling system. The MFT code and its additional guidance can be found at https://www.coaps.fsu.edu/~bourassa/MFT_html/MFT_docs.php (accessed on 1 May 2023). Input and output variables and options are provided in Appendix A.

### 3.2. Experimental Setup for Estimating Stress Changes Due to Oil-Related Changes

In this study, we calculated the changes in surface stress due to oil-related changes in momentum surface roughness, surface winds, and ABL stability using the flux model described in Section 3.1. We estimated the stress changes in four cases, which are listed in Table 2.

**Table 2.** Description of how the changes in surface stress were computed due to oil-related changes in surface roughness only, surface wind only, and boundary-layer stability only.

| Cases | Assumption for Oil-Related Parameters | Surface Stress Change |
|---|---|---|
| Case 1: | $z_{0m}$ is the only varying factor for partial oil coverage ($A_{oil} \neq 0$) | $\tau_{A_{oil} \neq 0} - \tau_{A_{oil} = 0 \approx water}$ |
| Case 2: | $z_{0m}$ is the only varying factor for total oil coverage ($A_{oil} = 1$) | $\tau_{A_{oil} = 1} - \tau_{water}$ |
| Case 3: | Surface wind is the only varying factor for partial oil coverage | $\tau_{A_{oil} \neq 0} - \tau_{A_{oil} \neq 0;\ water-related\ wind}$ |
| Case 4: | ABL stability is the only varying factor for partial oil coverage | $\tau_{A_{oil} \neq 0} - \tau_{A_{oil} \neq 0;\ water-related\ stability}$ |

$\tau_{A_{oil} \neq 0}$ is the surface stress for partial oil coverage; $\tau_{A_{oil} = 1}$ is the surface stress for the total oil coverage; $\tau_{water}$ is the surface stress for pure water coverage; $\tau_{A_{oil} \neq 0;\ water-related\ wind}$ is the surface stress for partial oil coverage with pure water-related surface wind; $\tau_{A_{oil} \neq 0;\ water-related\ stability}$ is the surface stress for partial oil coverage with pure water-related boundary-layer stability.

In case 1, the surface stress changes due to surface roughness changes were computed only between the partial oil coverage ($A_{oil} \neq 0$,) and the pure water assuming surface wind and boundary-layer stability were the same and from the non-oil simulation output; note that $A_{oil}$ was estimated in the oil simulation. Case 2 is the same as case 1 except it estimated the stress change using $A_{oil} = 1$ to compute surface roughness. In case 3, the surface stress changes caused by the surface wind changes were computed only between the partial oil coverage ($A_{oil} \neq 0$) and the pure water assuming surface roughness and boundary-layer stability were the same and from the non-oil simulation output. In case 4, the surface stress changes due to boundary-layer stability changes were computed only between the partial

oil coverage and the pure water assuming surface roughness and surface wind were the same and from the non-oil simulation output.

## 4. Results

### *4.1. Surface Stress Changes Due to Oil-Induced Changes in Surface Roughness*

To investigate how the discrepancy in surface roughness between oil and water surfaces contributes to changes in surface stress, we compared the changes in the magnitude of the surface stress when roughness is the only varying factor. Wind speed and boundary-layer stability are from the model run without oil. Figure 4 shows the probability distribution function (PDF) of the changes in surface stress magnitude due to the surface roughness differences between surface oil coverage and water surface coverage obtained in case 1 (Figure 4a) and case 2 (Figure 4b) over two weeks from 21 April 2010 to 5 May 2010, where case 1 is based on our estimated value of $A_{oil}$, and case 2 uses $A_{oil} = 1$. The surface stress changes were computed for oil locations only.

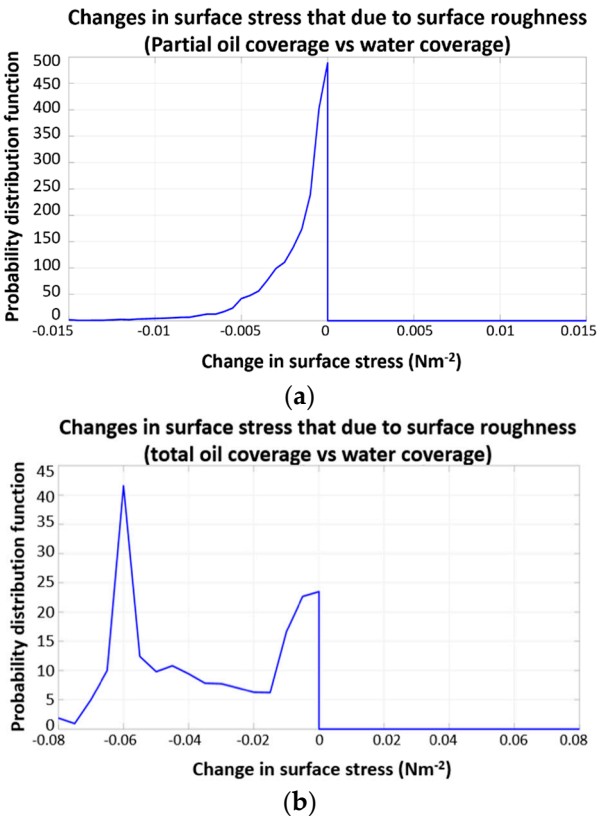

**Figure 4.** Probability distribution function of changes in surface stress (N m$^{-2}$) when roughness is the only varying factor while wind speed and boundary-layer stability are not impacted by oil over the 2-week period (21 April 2010–5 May 2010) for (**a**) partial oil coverage surface roughness relative to a water surface and (**b**) for total oil coverage surface roughness relative to a water surface. The negative changes in stress terms are from the conditions where roughness is suppressed by oil. There were no increases in stress solely due to oil impacting the roughness length.

In case 1, surface stress decreased when oil impacted the surface roughness since surface water waves can be suppressed by the partial coverage of an oil slick, leading to a small amount of surface roughness. However, the changes in surface stress magnitude only had a small dynamic range, indicating the changes in surface roughness due to the model-simulated oil coverage did not significantly impact surface stress. This probably occurred because there were substantial fractions of the surface that had no layer of oil (or the layer was so thin that it had little impact) despite there being plenty of oil slicks during the DWH spill period [44]. In case 2, when the water surface was completely covered by

oil, the changes in the magnitude of the surface stress were found to be much larger than in case 1. This indicates that a water surface completely covered by oil has a much larger impact on surface stress than an area with substantial gaps in oil coverage (i.e., patchy coverage). Since observations of the Deepwater Horizon spill indicated that the oil coverage was patchy [44], this simple first test reveals that the proper estimation of fractional oil coverage is likely to be necessary to model changes in air–sea interactions and surface oil motion transport.

### 4.2. Surface Stress Changes Due to Oil-Induced Changes in Surface Winds

Similar to Section 4.1, we examined the changes in the magnitude of surface stress due to changes in surface wind caused by the difference between water and oil alone. Both stresses were calculated with oil-free values of roughness length and boundary-layer stability. If the change in wind was due only to oil-induced changes in the roughness length, then we expected the reduced friction to increase the wind speed. Figure 5 shows the PDF of the changes in the magnitude of the surface stress for an oil-related 10 m wind relative to a water-related 10 m wind (case 3) over the same period as in Figure 4. The results show that the changes in surface wind due to the presence of an oil slick can increase or decrease surface stress. This happens because the surface wind speed can increase or decrease regardless of the presence of oil. This was expected because the changes in surface wind speed are not fully determined by the effect of oil alone but rather by other dominant processes (e.g., atmospheric-boundary-layer stability, changes in advection) that may come into play. Figure 6 demonstrates both the increases and decreases in the modeled surface speed of the oil spill region on 4 May 2010. Interestingly, the changes in surface stress magnitude due to surface wind changes appear to be larger than those due to the oil-related changes in surface roughness alone, again supporting the suggestion that factors beyond roughness length impact wind speed and stress.

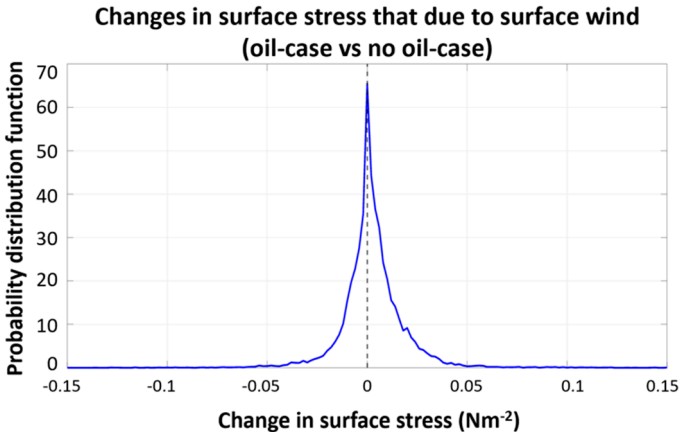

**Figure 5.** Probability distribution function of the changes in surface stress (N m$^{-2}$) for an oil-related 10 m wind relative to a water-related 10 m wind when the surface wind was the only varying factor over the 2-week period. On the left side of the curve are 34.98% of the data, and on the right side, 51.92% of the data are above the zero line.

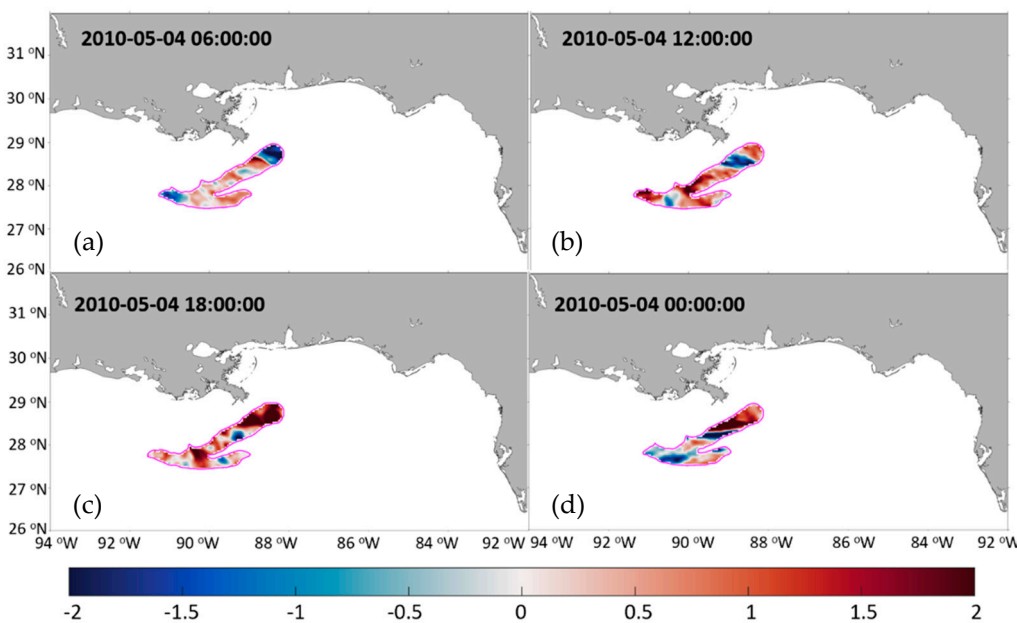

**Figure 6.** The differences in surface wind speed between the COAWST model with oil modifications relative to the COAWST model without oil modifications (shaded contours, m s$^{-1}$) on 4 May at (**a**) 06:00, (**b**) 12:00, (**c**) 18:00, and (**d**) 00:00 UTC.

*4.3. Surface Stress Changes Due to Oil-Induced Changes in Atmospheric-Boundary-Layer Stability*

Before we quantify the significance of changes in surface stress due to oil-induced changes in boundary-layer stability (i.e., dynamic stability at the base of the boundary layer), we will examine the PDF of the magnitude of air–sea potential temperature differences (i.e., the air potential temperature at a height of 2 m minus the SST) as a function of fraction oil coverage ($A_{oil}$). Figure 7 displays such temperature differences over the 2-week period (21 April 2010 to 5 May 2010) but only for 6 pm time-steps. The 6 pm time-steps were used because thick oil appears warmer than water during the daytime due to thermal isolation from the underlying water and the high absorption of solar radiation [45]. That time of day is presumably associated with a greater impact on air–sea temperature differences, thus highlighting oil-induced air–sea temperature differences in the atmospheric-boundary-layer stability. The results reveal that air–sea temperature differences can be positive or negative and strongly depend on the amount of oil coverage. A greater likelihood of negative air–sea temperature differences occurs over large surface oil coverages (e.g., the $0.1 < A_{oil} < 0.2$ and $A_{oil} > 0.2$ cases in Figure 7), implying conditions that are more likely to be thermodynamically unstable. In the absence of other considerations, more stable conditions (i.e., the change in the air–sea temperature difference is positive) tend to cause the 10 m wind and stress to be smaller, while more unstable air–sea temperature differences (i.e., the change in the air–sea temperature difference is negative) will tend to increase the 10 m wind and stress. It should be noted that the temperature changes could also be attributed to changes in oceanic and atmospheric horizontal heat advection.

Figure 8 shows the PDF of the change in the magnitude of surface stress for oil-related atmospheric-boundary-layer stability relative to water-related ABL stability over the 2-week period (21 April 2010 to 5 May 2010) for all 6 pm time-steps, as performed in case 4. These calculations used roughness length and wind speed from the model run without oil. The results reveal that a slick can increase or reduce surface stress due to oil-induced boundary-layer stability changes alone. The shape of the PDF is similar to that of the oil-induced surface wind change alone, as shown in Figure 5. Note that the changes in surface stress magnitude due to changes in oil-related surface wind alone (as in Figure 5) and oil-related ABL stability alone (as in Figure 8) are more likely larger than those due to changes in oil-induced roughness alone (as in Figure 4). Interestingly, most of the changes in surface stress magnitude fall near

and within the zero range (i.e., the stress change is still small), as seen in Figures 5 and 8, suggesting that the changes in boundary-layer stability and surface wind due to oil do not have a major impact on surface stress. These small changes in stress due to oil-induced processes are believed to cause small changes in the currents relative to the changes in currents caused by other effects such as intrinsic ocean dynamics. As a result, an oil-induced change in stress can have a minor impact on surface oil transport.

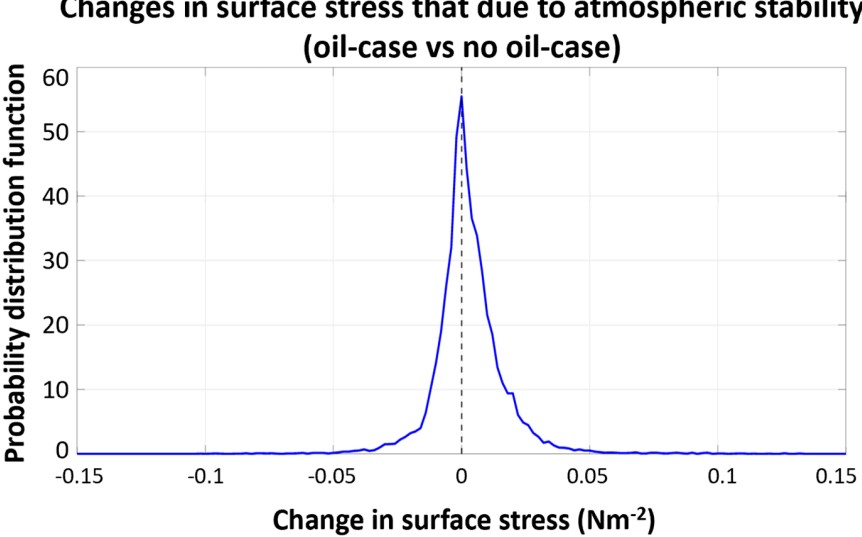

**Figure 7.** Probability distribution function of the changes in air–sea temperature differences for the COAWST model with and without oil modifications over a 2-week period (21 April 2010 to 5 May 2010), only in 6:00 p.m. time-steps for different $A_{oil}$ ranges.

**Changes in surface stress that due to atmospheric stability (oil-case vs no oil-case)**

**Figure 8.** Probability distribution function of the changes in surface stress (N m$^{-2}$) for oil-related boundary-layer stability relative to water-related boundary-layer stability when atmospheric-boundary-layer stability is the only varying factor over a 2-week period.

### 4.4. Relative Contributions to Surface Stress Changes Due to Oil-Induced Changes in Surface Roughness, Surface Wind, and Atmospheric-Boundary-Layer Stability

To compare the relative contributions to surface stress changes from the oil-induced changes in surface roughness, surface wind, and boundary-layer stability, we examined the PDF of percentage changes in the above three fields contributing to changes in surface

stress over the 2-week period (Figure 9). The percentage change in surface stress that is due purely to the changes in surface roughness, changes in surface wind, and changes in atmospheric-boundary-layer stability is defined as

$$Perc = \frac{(\overline{\tau_{oil}} - \overline{\tau_{no-oil}})}{\overline{\tau_{no-oil}}} \times 100,$$

where $\overline{\tau_{oil}}$ is the mean surface stress output that is calculated using variables from the oil-case experiment, and $\overline{\tau_{no-oil}}$ is the mean surface stress output that is calculated using the same variables from the oil-case experiment, except that the only varying factor (e.g., either surface roughness, surface wind, or ABL stability) is obtained from the no-oil case experiment. The contribution range purely from oil-induced surface roughness changes was the smallest (−4% to −1%) with the negative sign, suggesting it always decreases stress very slightly. The contribution purely from oil-induced surface wind ranges from −9% to +12%, and the contribution purely from oil-induced boundary-layer stability falls between −18% and 13%. While the change in roughness can have the greatest mean impact, changes in stress due to wind speed and stability can have much larger (albeit often partially canceling) impacts.

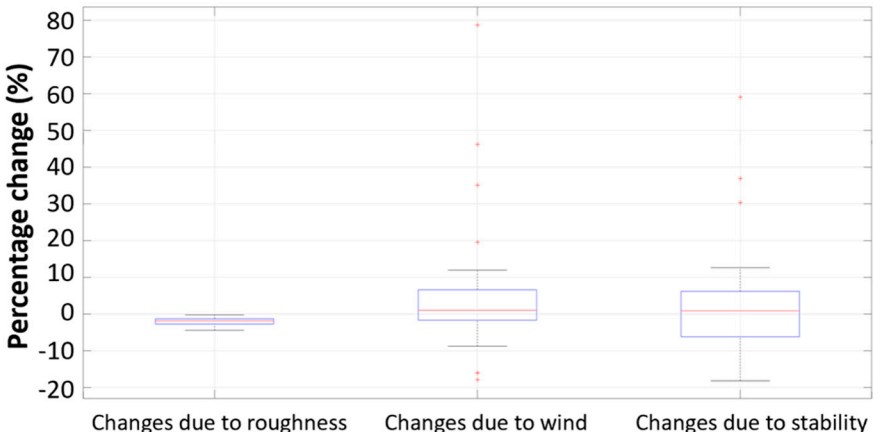

**Figure 9.** Box plots of percentage changes representing how much stress magnitude changes due to oil-induced changes in surface roughness only, surface wind only, and boundary-layer stability only. The length of the boxes (in blue) represents the interquartile ranges, the median is represented by the middle line (in red), and the lengths of the whiskers (in black) indicate the scatter of data away from the quartile ranges. The calculation is based on the prescribed 2-week period.

Note that the contributions from each of the three fields are not independent. For example, surface roughness changes due to oil can substantially affect wind speed, and both of these variables impact wind stress and the ocean surface current. Furthermore, the presence of oil itself could affect wind speed and boundary-layer stability via changes in air–sea temperature differences and thus could contribute to an increase in stress. The interwoven features above can be further quantitatively illustrated in Figure 10. It can be seen that the changes in stress due to wind are very well correlated with the changes in the surface due to boundary-layer stability (Figure 10). This is not surprising because increases in wind and the SST enhance stress. This suggests most of the changes in the modeled stress were due to the changes in the modeled stability coupled with the changes in the modeled wind. This indicates that friction velocity plays a large role in the driving physics factor that is influenced by oil and thus affects air–sea interaction and currents, but this also strongly suggests that oil-induced changes in the SST are more important than oil-induced changes in roughness length.

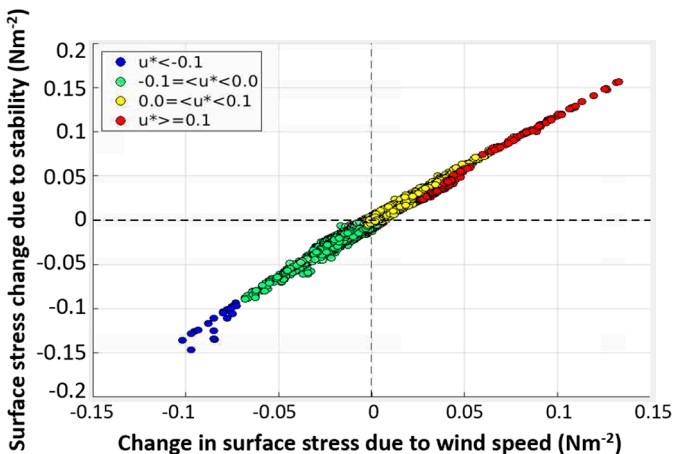

**Figure 10.** Scatterplot of changes in surface stress due to boundary-layer stability vs. changes in stress due to wind speed, showing a function of friction velocity (with different colors) over the prescribed 2-week period. Unit of stress: N m$^{-2}$.

## 5. Discussion

Oil slicks cause changes in surface stress and boundary-layer stability over areas of high oil coverage and low wind speeds (<8 m/s). If wind exceeds 18 m/s, the oil is likely to be dispersed into the water column [46], and it has a limited ability to dampen the ocean surface waves [47]; thus, forecasting an oil spill within a high wind environment is relatively easy. This study focused on the effects of oil on stress within a low-to-moderate wind environment. Our results indicate that oil-induced differences in boundary-layer stability contribute to changes in surface stress for conditions typical of the DWH spill (i.e., light and moderate winds). The boundary-layer stability is largely dependent on wind speed, SST, and surface air temperature. Our results show that the changes in air–sea temperature differences are relatively important and should be considered in future modeling.

Fractional oil coverage is important in making changes in air-sea temperature differences; hence, it modifies boundary-layer stability (stable and unstable). The drag coefficient is a crucial parameter in quantifying stress over the ocean surface, which depends on ocean surface roughness. Figure 11 demonstrates how the five different fractional oil coverages ($A_{oil}$ = 0.0, 0.25, 0.50, 0.75, and 1.0) influence boundary-layer stability and surface stress (via the drag coefficient) for different ranges of air–sea temperature differences and 10 m wind speed conditions. When the air potential temperature is greater than the SST (green lines), the atmosphere is considered stable, resulting in less surface stress. Conversely, when the air potential temperature is less than the SST (red lines), the atmosphere is considered less stable, resulting in more significant surface stress. The changes in stress corresponding to a 1 °C change in the air–sea temperature difference are much greater for stable conditions than for unstable conditions, regardless of the fractional oil coverage. The peak of drag coefficients between 2 and 4 m s$^{-1}$ is due to capillary waves (wavelength < 1.7 cm) and is reduced as the fractional oil coverage increases. The local minima in drag coefficients indicate that capillary waves and gravity waves (wavelength > 1.7 cm) make equal contributions to the roughness. The roughness due to capillary waves decreases as wind speed increases, whereas the roughness due to gravity waves increases as wind speed increases. As the amount of oil increases, the minima in drag coefficients move to a greater wind speed, indicating that the presence of oil dampens both capillary waves and short gravity waves. Figure 11 shows that when oil covers the ocean surface, the oil changes the drag coefficient. As the fraction of oil coverage increases, the drag coefficient decreases. The above statement explains why surface oil roughness can decrease the surface stress and surface current if the oil roughness is the only dominant factor. These figures show qualitative agreement with Figure 4b, in that the amount of oil coverage can affect the wind stress and current, which affects the oil's movement. The effects of oil on changes in

boundary layer processes and oil motion have not been included in oil spill models. The results of this study indicate that this could be useful in considering the effect of oil changes on the atmospheric/oceanic process, which can impact the movement of the oil, especially in large-scale oil spill situations.

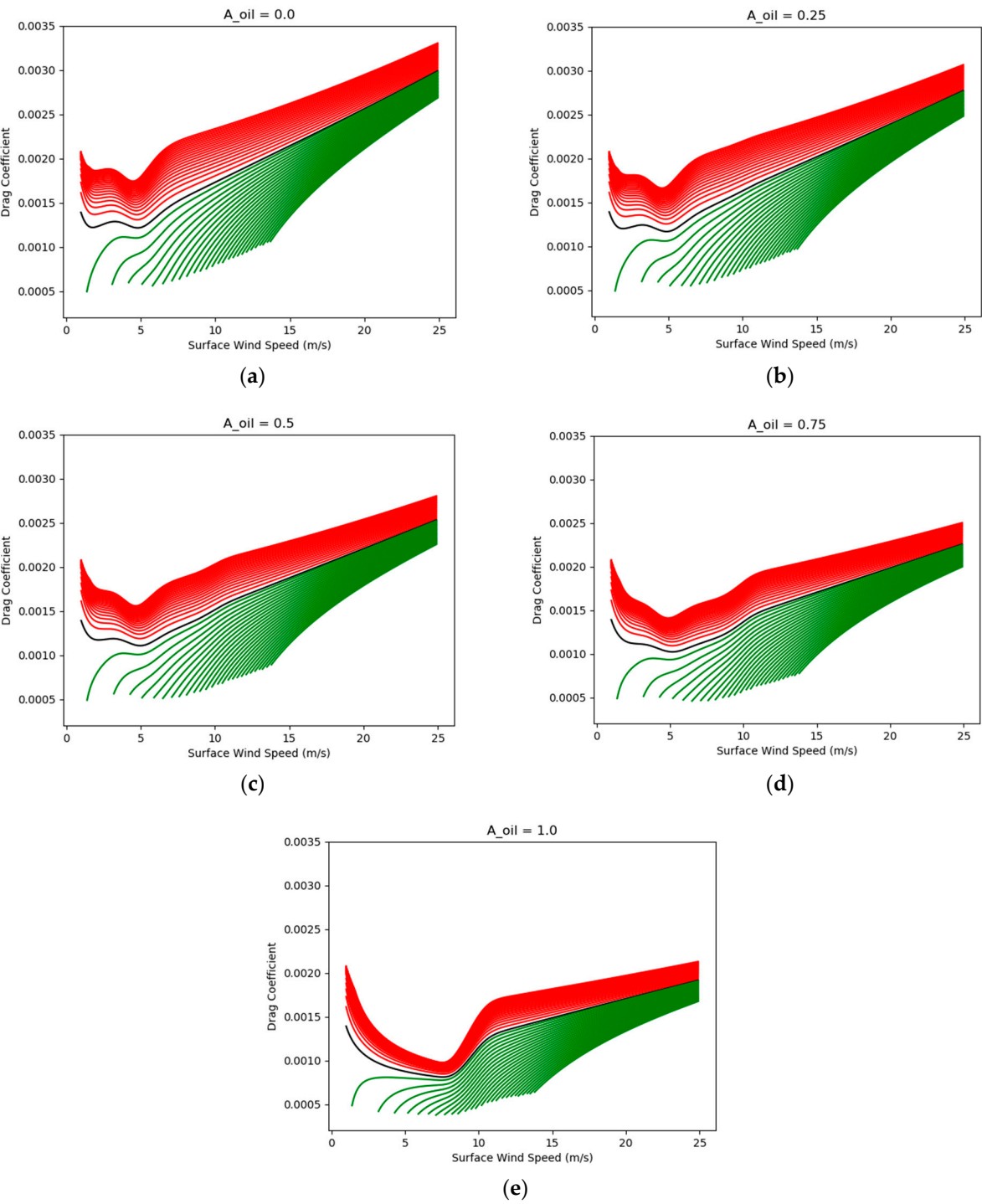

**Figure 11.** Variations in drag coefficients for air–sea temperature differences of −40 °C to +25 °C and wind speeds of 10 m in height for fractional oil coverage values of (**a**) 0.0, (**b**) 0.25, (**c**) 0.50, (**d**) 0.75, and (**e**) 1.0. The red lines represent when the air temperature is less than the SST, the green lines represent when the air temperature is greater than the SST, and the black line represents when the air temperature is equal to the SST. Red lines are every 2 °C, whereas green lines are every 1 °C.

The major uncertainty in our results (i.e., ignoring the weathering of oil) is the dependencies of the atmospheric/oceanic processes on the quantity of oil. Uncertainty in the parameterization of boundary-layer processes (e.g., roughness length, stability, and the impacts of stability) makes it complex to examine the effects of oil separately on surface roughness, surface wind speed, and boundary-layer stability and how those conditions impact surface stress. Here, we used only the roughness parameterization adapted from [7] and stability impacts from the WRF. A larger fraction of coverage would result in more significant changes in stress and the transport of the oil, and a greater sensitivity to oil would also increase the importance of considering oil slicks with a coupled model. Furthermore, oil may increase the wind speed needed for substantial large-scale wave breaking; however, the SWAN model uses input winds to model stress, so we could have more wave breaking when there should be less wave breaking.

Figure 12 presents a schematic diagram of the feedback mechanism of oil slick and air–sea coupling. The main dominant fact that the oil slick impacted was friction velocity (see Figure 10). The left-hand side of Figure 12 shows how oil coverage modifies roughness length and, hence, friction velocity, surface wind, and boundary-layer stability. All these variables depend on friction velocity; thus, this makes oil and air–sea coupling more complicated. The right-hand side of Figure 12 shows how fractional oil coverage modifies the air–sea interaction through temperature differences. An oil slick can reduce the latent heat flux by inhibiting evaporation. A reduction in the latent heat flux increases the SST, which will also warm the air.

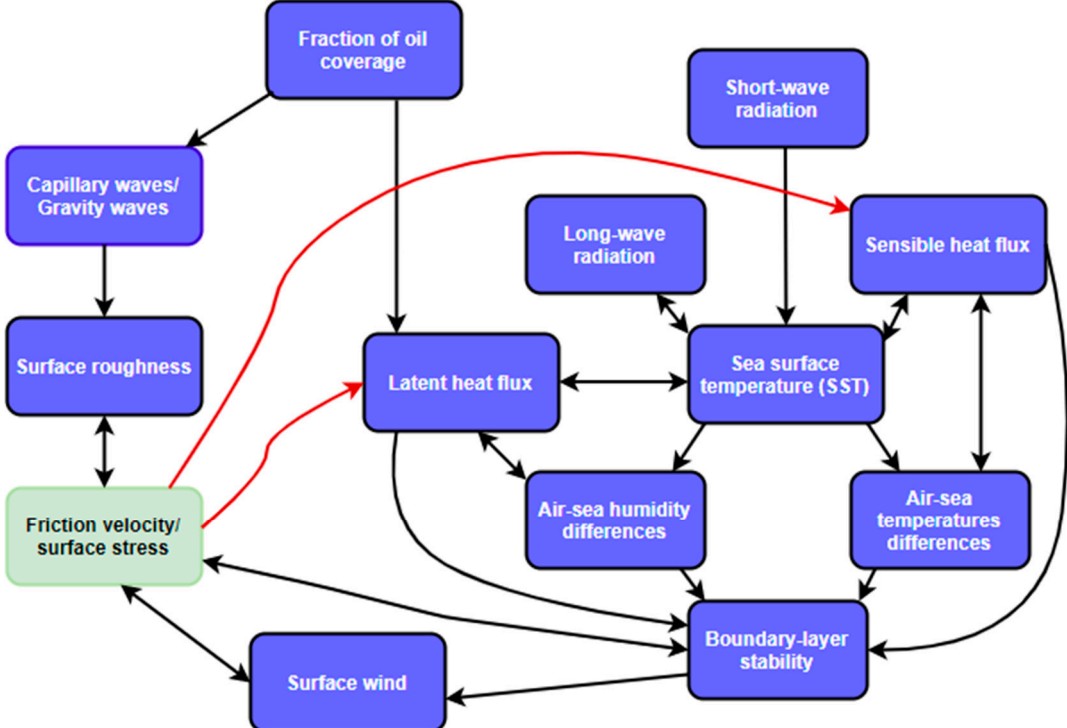

**Figure 12.** A schematic diagram of the feedback mechanism indicating how an oil slick impacts air–sea coupling. The dominant factors for the impact of the oil slick are surface roughness and SST. Oil affects surface roughness, which impacts surface stress. Oil also affects the SST, causing a change in the air–sea temperature difference, leading to a change in boundary-layer stability and ultimately impacting the surface stress. Oil-induced changes in friction velocity and boundary-layer stability also cause changes in surface wind, which impacts surface stress.

## 6. Summary

This study provides the first detailed analysis of how oil changes air–sea interactions in a two-way coupled model. Specifically, we focused on exploring how oil modifies surface

roughness, surface wind, and boundary-layer stability and how these oil-induced changes affect surface stress separately. We chose the DWH oil spill in the Gulf of Mexico from 20 April to 5 May 2010 as a simulation scenario where a high wind environment is excluded. Three major conclusions were made:

(1) Oil-related changes in surface roughness are not significant enough to cause a large impact on surface stress changes;

(2) Oil-related changes to 10 m wind speed and boundary-layer stability have a relatively greater impact (albeit often partially canceling) than oil-related surface roughness changes on surface stress changes, though the change in roughness has the greatest mean impact;

(3) Oil-related changes in surface stress are not large enough to cause a major change in the ocean currents as compared with other effects, such as intrinsic ocean dynamics; thus, oil-induced changes in stress have a limited impact on surface oil transport.

The above results are subject to uncertainties in the amount of oil coverage and the parameterizations for boundary-layer processes. Air–sea temperature differences that affect boundary-layer stability can also be modified by atmospheric advection, which was not examined in this study. This study reveals that the influence of oil-related temperature and oil-related surface roughness for oil trajectory forecasting is minor due to the small impact on surface stress unless the fractional oil coverage is much larger than the value found in this study. Thus, it could be useful to consider the effect of oil-related changes on atmospheric/oceanic processes, which can impact oil transport, especially in large-scale oil spill situations. Sensitivity to the fractional coverage of oil is demonstrated in several ways and could be underestimated in this modeling study. If it is greatly underestimated, the conclusions about the impact of oil on stress and currents might be changed. This study produced tools and a methodology that are well suited to testing these conclusions when and if better oil coverage data are available.

**Author Contributions:** Conceptualization, Y.Z. and M.A.B.; methodology, Y.Z., D.B. and M.A.B.; validation, D.B., Y.Z. and M.A.B.; formal analysis, D.B.; investigation, D.B.; data curation, D.B.; writing—original draft preparation, D.B.; writing—review and editing, Y.Z. and M.A.B.; visualization, D.B.; supervision, Y.Z. and M.A.B.; project administration, Y.Z.; funding acquisition, Y.Z. and M.A.B. All authors have read and agreed to the published version of the manuscript.

**Funding:** This research was funded by the Gulf of Mexico Research Initiative (GoMRI) RFP-VI, grant number G-231820, and the U.S. National Aeronautics and Space Administration (NASA) Physical Oceanography of the Ocean Vector Winds Science Team (OVWST), subcontract number 1639996.

**Data Availability Statement:** All datasets used in this study are available at https://doi.org/10.5281/zenodo.7826442, accessed on 13 April 2023.

**Acknowledgments:** Model simulations for this study were conducted on the HPC at the Florida State University Research Computing Center (FSU RCC), and data analysis was performed at the Center for Ocean-Atmospheric Prediction Studies (COAPS) at FSU. We thank the COAWST modeling system for open access to their codes and their technical support. Daneisha Blair was supported by the GoMRI RFP-VI; Yangxing Zheng and Mark A. Bourassa were supported by the GoMRI RFP-VI and the U.S. NASA OVWST.

**Conflicts of Interest:** The authors declare no conflict of interest.

**Appendix A**

Parameterization options available within the Modularized Flux Testbed are listed in the table below. Options can be selected independently for momentum, temperature, and moisture parameterizations. The selections used in this study have a light gray background. In the columns for stability adjustment parameterizations, the parametrization above the dashed line is for unstable conditions, and the parameterization below the dashed line is for stable conditions.

| Option # | Momentum Roughness Length | Temperature and Moisture Roughness Length | Momentum Stability Adjustment | Temperature and Moisture Stability Adjustments |
|---|---|---|---|---|
| 0 | Bourassa, Vincent, and Wood (BVW [31]) | Aerodynamically smooth surface | Benoit's (1977, [48]) adaption of Dyer (1974, [33]) ——————— Beljaars and Holtslag (1991, [49]) | Benoit's (1977, [48]) adaption of Dyer (1974, [33]) ——————— Beljaars and Holtslag (1991, [49]) |
| 1 | Bourassa (2006, [30]) | Clayson, Fairall, and Curry (1996, [50]) | Dyer 1974, [33]) ——————— Hicks (1976, [34]) | Dyer (1974, [33]) ——————— Hicks (1976, [34]) |
| 2 | Taylor and Yelland (2001, [51]) with BVW capillary wave roughness | Zilitinkevich et al. (2000, [52]) | Benoit's (1977, [48]) adaption of Dyer (1974, [33]) ——————— Hicks (1976, [34]) | Benoit's (1977, [48]) adaption of Dyer (1974, [33]) ——————— Hicks (1976, [34]) |
| 3 | Taylor and Yelland (1999, [51]) | Liu, Katsaros and Businger (1979, [53]) | Dyer (1974, [33]) ——————— Hicks (1976, [34]) with a solution for lower boundary conditions | Dyer (1974, [33]) ——————— Hicks (1976, [34]) with a solution for lower boundary conditions |
| 4 | Zheng et al. (2013, [7]) | COARE 3.0 Fairall et al. (2003, [54]) | | |
| 5 | Aerodynamically smooth surface | Griffin (2009, [55]) retuned CFC | | |
| 6 | Oil spill parameterization (this study) | | | |
| 7 | Input a value | | | |

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
