# Peer review of "The Effect of Surface Oil on Ocean Wind Stress"

_2673-4834, doi:10.3390/earth4020019_

Round 1
Reviewer 1 Report
The present study examines the impact of surface oil on ocean wind stress.
Regarding line 38, could you kindly explain why gravity initially causes the rapid spreading of oil?
On line 49, would you please enumerate the limitations of these methods?
For lines 65 and 66, you may want to reference "Xiao, S., & Yang, D. (2020). Effect of oil plumes on upper-ocean radiative transfer—A numerical study. Ocean Modelling, 145, 101522," which is a recent study relevant to this topic.
In equation 10, is z_\theta modeled in the same way as z_mom? Additionally, could you provide the explicit formula for the relationship between oil coverage and SST? It would be intriguing for readers to understand how varying oil coverage affects SST positively or negatively, as demonstrated in Figure 7.
Considering you've only included oil coverage, is it possible that oil concentration could also play a role in this study?
In Figure 9, the colors are not easily distinguishable. Based on the mean, it appears that the contribution of roughness is larger compared to other factors, while the variations for other factors are much larger. Therefore, your assertion that the impact of roughness change due to oil slick is minimal may not be accurate.
On line 508, could you please explain why strong winds would push the oil slick deep into the water column?
In line 532, you mentioned the local minima from Figure 11, which is not entirely clear to me. Would you mind elaborating further?
Regarding lines 590-594, your assessment of the relative importance between roughness and wind may not be entirely accurate. Please refer to the previous comments for clarification.
Lastly, what is the change in roughness due to the presence of an oil plume? This information would likely be of interest to those studying atmospheric boundary layer (ABL) flow.
Author Response
Response to Reviewer 1
We greatly appreciate the reviewer’s helpful comments on this manuscript. Responses to specific comments are given below (italic text indicates the reviewer’s original comments).
The present study examines the impact of surface oil on ocean wind stress.
(1). Regarding line 38, could you kindly explain why gravity initially causes the rapid spreading of oil?
Answer: When oil first reaches surface, there are abundance of oil accumulated over a small area of water surface. These quick accumulated crude oils from below leading to a horizontal gradient in density. Oil is very non-viscous and will spread rapidly as a density flow which is affected mainly by gravity and density of the two fluids.
(2). On line 49, would you please enumerate the limitations of these methods?
Answer: For example, none of these models incorporated Stokes drift. They also did not consider biological consumption of oil or the physical-chemical weathering processes. We will include the limitations of these methods in the revised manuscript (Lines 50–52).
(3). For lines 65 and 66, you may want to reference "Xiao, S., & Yang, D. (2020). Effect of oil plumes on upper-ocean radiative transfer—A numerical study. Ocean Modelling, 145, 101522," which is a recent study relevant to this topic.
Answer: Thanks. We have added the reference to the paper on line 68.
(4). In equation 10, is z_\theta modeled in the same way as z_mom? Additionally, could you provide the explicit formula for the relationship between oil coverage and SST? It would be intriguing for readers to understand how varying oil coverage affects SST positively or negatively, as demonstrated in Figure 7.
Answer: z0 for momentum and for temperature are treated differently (see Appendix A in the manuscript). There is no explicit formula for the relationship between Aoil and SST because SST is not solely determined by fractional oil coverage (i.e., Aoil). But based on Figure 7, the large Aoil can be more likely to produce negative air temperature-SST differences, with a most common change of 0.005 K for Aoil = 10% and 0.01K for Aoil >20%. Our study does not have large values of Aoil, which would presumably have much larger changes in air/sea temperature differences. Since the air is also warming, the change in SST would be larger than change in the air/sea difference.
(5). Considering you've only included oil coverage, is it possible that oil concentration could also play a role in this study?
Answer: The answer is “Yes” because in this study the fractional oil coverage (i.e., Aoil) is taken to be a linear function of surface oil concentration (see section 2.4). If the oil were more viscous it would be more concentrated and would have a greater impact on stress. We note as a weakness that our impact of oil is tuned to DWH oil.
(6). In Figure 9, the colors are not easily distinguishable. Based on the mean, it appears that the contribution of roughness is larger compared to other factors, while the variations for other factors are much larger. Therefore, your assertion that the impact of roughness change due to oil slick is minimal may not be accurate.
Answer: Thank you for this correction. We have fixed our statement to be clear that it refers to mean impact. We have replaced ' It appears that contributions from surface wind and stability changes are more likely larger than that purely from surface roughness’ with ‘While the change in roughness has the greatest mean impact, changes in stress due to wind speed and stability can have much bigger (albeit often partially cancelling) impacts.' (Lines 468–470).
(7). On line 508, could you please explain why strong winds would push the oil slick deep into the water column?
Answer: We say 'into the water column' rather than 'deep into the water column.' Wave breaking causes mixing of the oil and water, provided there is sufficient wind stress to generate wave breaking. Breaking re-emulsifies the oil into small droplets. Typically, they will not mix very far down. They can also return to the surface after the high wind event. Something similar is seen with plastics.
(8). In line 532, you mentioned the local minima from Figure 11, which is not entirely clear to me. Would you mind elaborating further?
Answer: At very low wind speeds, the stress is dominated by viscous processes associated with a smooth surface, which have a drag coefficient that goes to zero as wind speed approaches zero. When winds are well represented by a mean wind speed (i.e., in a wave tank) as the wind speed increases to about 1.8 m/s, capillary waves will form. In ocean conditions, capillary waves form over only a portion of the surface because they reduce the wind speed where there are capillary waves. The wind speed is distributed, supporting capillary waves over a greater fraction of the surface as the wind speed increases. Capillary waves also have a drag coefficient that approaches infinity as the wind speed approaches zero, and this drag coefficient is larger than that for a smooth surface. However, capillary waves are not present for very low wind speeds, resulting in the first local minima in Fig. 11a. The second local minima is located where the roughness from capillary waves equals the roughness from gravity waves. Gravity wave roughness increases with increasing wind speeds.
(9). Regarding lines 590-594, your assessment of the relative importance between roughness and wind may not be entirely accurate. Please refer to the previous comments for clarification.
Answer: We agree that the statement must be more carefully worded (see details in our response to question 6). We have made similar corrections in the conclusions (Lines 586–588).
(10). Lastly, what is the change in roughness due to the presence of an oil plume? This information would likely be of interest to those studying atmospheric boundary layer (ABL) flow.
Answer: That is an interesting question! If the plume (the subsurface oil) could impact the ocean currents, then it could have an impact on the roughness and stress. We are not using a physical oceanographic representation of the plume, so we can't access that question with this model. Under most conditions we speculate that the impacts on roughness and stress are small. However, there might be greater impact after a storm when the oil might cause a vertical gradient in density, which could trap more of the wind-induced momentum near the surface. This would likely be a short-lived impact because it is time-limited between the time when large scale breaking isn't important and the time when it takes the oil to return to the surface.
Reviewer 2 Report
This manuscript presents a numerical modelling study of the influence of surface oil on wind-wave interactions. The authors conducted well-designed, controlled numerical simulations to figure out the impact of surface oil on the change of surface stress, wind speed and atmospheric boundary layer stability. The analysis is clearly performed. The results are of interest to the field, considering that there is a critical need to understand the physical impact of marine oil spill events on the ocean and atmosphere. I would like to suggest its publication. Below are some suggestions and comments, hoping to improve the quality of this manuscript.
1. In the introduction, besides the RANS modelling, the authors should also review the recent development in the two-way coupled DNS approach for simulating air-sea interactions. For example, see Komori et al. JoT 2010; Li & Shen JFM 2022; Cimarelli et al. JFM 2023.
2. Oil can also impact the dynamics of wave-breaking statistics and modify the wave dissipation properties, which may alter the dissipation term in the SWAN model. It is suggested that the authors include a brief discussion about the impact of surface oil on the wave-breaking effect.
3. In Section “4.3. Surface Stress Change due to Oil-induced Change in Atmospheric Boundary-Layer Stability”, it would be better to clarify which kind of atmospheric boundary layer is of interest to study, e.g., the stable atmospheric boundary layer, or the convective (unstable) atmospheric boundary layer.
4. In the reference [18], the user manual of WRF Version 2 is cited. To my knowledge, WRF Version 4 has been released. If the authors use a newer version of WRF, please consider updating the reference.
Reference:
· Komori, S., Kurose, R., Iwano, K., Ukai, T., & Suzuki, N. (2010). Direct numerical simulation of wind-driven turbulence and scalar transfer at sheared gas–liquid interfaces. Journal of Turbulence, (11), N32.
· Li, T., & Shen, L. (2022). The principal stage in wind-wave generation. Journal of Fluid Mechanics, 934, A41.
· Cimarelli, A., Romoli, F., & Stalio, E. (2023). On wind–wave interaction phenomena at low Reynolds numbers. Journal of Fluid Mechanics, 956, A13.
Author Response
Response to Reviewer 2
We greatly appreciate the reviewer’s helpful comments on this manuscript. Responses to specific comments are given below (italic text indicates the reviewer’s original comments).
This manuscript presents a numerical modelling study of the influence of surface oil on wind-wave interactions. The authors conducted well-designed, controlled numerical simulations to figure out the impact of surface oil on the change of surface stress, wind speed and atmospheric boundary layer stability. The analysis is clearly performed. The results are of interest to the field, considering that there is a critical need to understand the physical impact of marine oil spill events on the ocean and atmosphere. I would like to suggest its publication. Below are some suggestions and comments, hoping to improve the quality of this manuscript.
(1). In the introduction, besides the RANS modelling, the authors should also review the recent development in the two-way coupled DNS approach for simulating air-sea interactions. For example, see Komori et al. JoT 2010; Li & Shen JFM 2022; Cimarelli et al. JFM 2023.
Answer: Good point and thanks! We have included a brief description of recent development in the two-way coupled DNS approach for simulating air-sea interaction and include them as important references for the recent development of two-way coupled simulations (line 92).
(2). Oil can also impact the dynamics of wave-breaking statistics and modify the wave dissipation properties, which may alter the dissipation term in the SWAN model. It is suggested that the authors include a brief discussion about the impact of surface oil on the wave-breaking effect.
Answer: The reviewer's suggestion is plausible. Oil may increase the wind speed needed for substantial large-scale breaking. Unfortunately, the SWAN model uses input wind to model stress, so we could have more breaking when there should be less breaking. We have added this discussion on lines 548-551.
(3). In Section “4.3. Surface Stress Change due to Oil-induced Change in Atmospheric Boundary-Layer Stability”, it would be better to clarify which kind of atmospheric boundary layer is of interest to study, e.g., the stable atmospheric boundary layer, or the convective (unstable) atmospheric boundary layer.
Answer: In this study, the ABL stability means either stable or unstable conditions due to changes in air-sea temperature differences. We have made it clear (lines 405–406) that stability refers to dynamic stability at the base of the boundary-layer.
(4). In the reference [18], the user manual of WRF Version 2 is cited. To my knowledge, WRF Version 4 has been released. If the authors use a newer version of WRF, please consider updating the reference.
Answer: The new version of WRF was not released for the COAWST modeling system when we conducted the modeling.
Reference:
- Komori, S., Kurose, R., Iwano, K., Ukai, T., & Suzuki, N. (2010). Direct numerical simulation of wind-driven turbulence and scalar transfer at sheared gas–liquid interfaces. Journal of Turbulence, (11), N32.
- Li, T., & Shen, L. (2022). The principal stage in wind-wave generation. Journal of Fluid Mechanics, 934, A41.
- Cimarelli, A., Romoli, F., & Stalio, E. (2023). On wind–wave interaction phenomena at low Reynolds numbers. Journal of Fluid Mechanics, 956, A13.
Reviewer 3 Report
The article is- The Effect of Surface Oil on Ocean Wind Stress
· Is there any reason for selecting oil spill period from April 21st to May 5th?
· Why did you select MFT physics parameterization?
· Could you please explain how the study applied the flux model?
· In the results, section 4.1, the article compares the changes in surface stress magnitude when roughness is the only varying factor. The results show that the changes in surface stress magnitude due to the model-simulated oil coverage do not significantly impact surface stress. How does this occur?
· Section 4.2- It examines the changes in the magnitude of surface stress due to the changes in surface wind alone caused by the difference between water and oil. The results show that the changes in surface wind due to the presence of an oil slick can increase or decrease surface stress. How does this happen?
· Can you explain more about how the flux model works and how it was used in this study?
· How were the parameterizations in the MFT selected to ensure consistent surface stress calculations with the output from the COAWST modeling system?
· How was the oil-related change in momentum surface roughness, surface winds, and ABL stability estimated and incorporated into the flux model calculations?
· What was the rationale for selecting the four cases used to estimate the surface stress changes due to oil-related changes in surface roughness, surface wind, and ABL stability?
· Were any assumptions made about the properties of the oil or the water in the simulations, and if so, how were these assumptions justified?
Author Response
Response to Reviewer 3
We greatly appreciate the reviewer’s helpful comments on this manuscript. Responses to specific comments are given below (italic text indicates the reviewer’s original comments).
The article is- The Effect of Surface Oil on Ocean Wind Stress
(1). Is there any reason for selecting oil spill period from April 21st to May 5th?
Answer: The scenario for this study is the Deepwater Horizon (DWH) blowout where oil continued to reach the surface (started on April 20, 2010) until the oil well-head was sealed off (ended on May 5, 2010). On April 20, the oil slick was initially treated as distinct from water and spread as a density flow on the surface and this initial expansion is very rapid (shorter than one hour). The effect of surface oil on stress is not evident due to its small oil coverage in the first day. However, the surface oil coverage is big enough to potentially affect stress since April 21st. Because this paper is focused on the impact of surface oil on stress, the analysis period starting on April 21st was selected when oil surface is big enough to impact stress.
(2). Why did you select MFT physics parameterization?
Answer: As mentioned in section 3.1, the differences in the outputs of twin coupled experiments are due to the combined oil-related changes in surface roughness, surface wind, and atmospheric stability, which does not directly allow us to tell which of oil-related changes contributes most to the surface stress changes. To partition the contribution of oil-related changes to the surface stress, we needed to be able to modify the code used to calculate fluxes in a manner that allowed us to change some variables but not others, including variables calculated by the model. That was easiest to do with code that was already modularized. These modifications were also consistent with the teaching needs of one of the coauthors.
(3). Could you please explain how the study applied the flux model?
Answer: The flux model is designed based on the equations (1) through (11). The flux model to estimate the wind stress is basically determined by the friction velocity (see eq (6) in the paper), and friction velocity is determined from eq (9), which are associated with the surface wind, surface current, ABL stability, and surface roughness length for momentum (eq (1)). Surface wind, surface current, and ABL stability are obtained from model outputs, and surface roughness length for momentum is also associated with the ocean state derived from the model outputs. The differences between oil and water surface produce different model outputs, which contribute to the changes in wind stress (estimated from flux model). Note these changes in wind stress are due to the combined oil-related changes in surface wind, surface roughness, and ABL stability estimated from the flux model embedded in the COAWST modeling system.
(4). In the results, section 4.1, the article compares the changes in surface stress magnitude when roughness is the only varying factor. The results show that the changes in surface stress magnitude due to the model-simulated oil coverage do not significantly impact surface stress. How does this occur?
Answer: For wind speeds over the Gulf of Mexico, stress is typically tiny. For a wind speed of 5 m/s, the stress is roughly 0.03 N/m2, and it is about 0.08 N/m2 for 8 m/s. The impact of surface oil on wind stress is determined by the modeled fractional oil coverage. Figure 4a used the modeled fractional oil coverage to estimate the wind stress and find minor impacts on surface stress. The minor impact is due to the patchiness of oil simulated in the model. The patchiness of oil coverage from the models is consistent with the observations. However, if we prescribed Aoil = 1.0 (which is not from modeled Aoil), which indicates a case when a water surface was completely covered by oil. Apparently, in this case, the impact of surface oil on wind stress (Figure 4b) is much larger than in Figure 4a when oil is patchy. Comparisons between Fig. 4a and 4b indicate the importance of accurately modeled oil fractional coverage in the change of surface stress.
(5). Section 4.2- It examines the changes in the magnitude of surface stress due to the changes in surface wind alone caused by the difference between water and oil. The results show that the changes in surface wind due to the presence of an oil slick can increase or decrease surface stress. How does this happen?
Answer: The increase or decrease surface stress due to oil-related surface wind change is expected. This happens because the surface wind speed can increase or decrease at the presence of oil (see Figure 6). In fact, the surface wind is not solely determined by the effect of oil but by other dominant factors such as ABL stability, atmospheric advection, and so on.
(6) Can you explain more about how the flux model works and how it was used in this study?
Answer: See explanations in (3).
(7). How were the parameterizations in the MFT selected to ensure consistent surface stress calculations with the output from the COAWST modeling system?
Answer: We added the roughness parameterizations used in the MFT as an option in WRF. The flux model is identical for MFT and for the COAWST modeling system so that the surface stress is calculated using the same method. The only purpose of using MFT (which include the flux model) is to partition the contribution of surface roughness, surface wind, and ABL stability to the surface stress with the model outputs, in which each of the varying factors (i.e., surface roughness, surface wind, and ABL stability, respectively) are obtained from the coupled models with and without oil effects included, while other factors to estimate surface stress are the same for oil case and no-oil case, which are provided only from the coupled model output without oil effect included.
(8). How was the oil-related change in momentum surface roughness, surface winds, and ABL stability estimated and incorporated into the flux model calculations?
Answer: As explained in (3), the flux model (equations (1)–(11)) considers the differences between oil and water surface in damping effect (i.e., surface roughness), in modulating surface wind (i.e., frictional drag), and in modification of ABL stability (i.e., caused by the SST-surface air temperature differences ( due to oil-related changes in SST, sensible and latent heat (water vapor evaporation) fluxes). The separate effect on surface stress due to oil-related changes in surface roughness, surface winds, and ABL stability were computed in the flux model as explained in (7). In many of our examples, this required that inputs to the model or values that flux models normally calculate were set rather than calculated.
(9). What was the rationale for selecting the four cases used to estimate the surface stress changes due to oil-related changes in surface roughness, surface wind, and ABL stability?
Answer: After initially running the twin experiments we found a great deal of variability in the response to oil and wanted to understand the cause of that variability. We selected 4 cases listed in Table 2 because we want to examine the importance of fractional oil coverage (case 1 and case 2) in affecting surface stress and because we want to investigate the relative importance of oil related changes in surface roughness, surface wind, and ABL stability in changing the surface stress (case 1, case 3, and case 4) since the oil-related changes in the above three factors can potentially result in changes in surface stress.
(10). Were any assumptions made about the properties of the oil or the water in the simulations, and if so, how were these assumptions justified?
Answer: In some cases, the properties of oil are known (such as the impact on evaporation or radiative properties of oil). In other cases, they are based on direct observations, such as the characteristics of the plume. The roughness length parameterization (Zheng et al. 2013) was based on observed changes in scatterometer backscatter. That backscatter is well known to be a function of stress and linked closely to short water waves. We simply tuned the stress to be more consistent with the buoy winds and scatterometer stress.
References appearing in the response letter:
Zheng, Y.; Bourassa, M.A.; Hughes, P. Influences of sea surface temperature gradients and surface roughness changes on the motion of surface oil: A simple idealized study. J. Appl. Meteor. Climatol. 2013, 52 (7), 1561–1575, https://doi.org/10.1175/JAMC-D-12-0211.1